# Catching Them Early: Framework Parameters and Progress for Prenatal and Childhood Application of Advanced Therapies

**DOI:** 10.3390/pharmaceutics14040793

**Published:** 2022-04-05

**Authors:** Carsten W. Lederer, Lola Koniali, Tina Buerki-Thurnherr, Panayiota L. Papasavva, Stefania La Grutta, Amelia Licari, Frantisek Staud, Donato Bonifazi, Marina Kleanthous

**Affiliations:** 1The Molecular Genetics Thalassemia Department, The Cyprus Institute of Neurology & Genetics, Nicosia 2371, Cyprus; lolak@cing.ac.cy (L.K.); panayiotap@cing.ac.cy (P.L.P.); marinakl@cing.ac.cy (M.K.); 2Empa, Swiss Federal Laboratories for Materials Science and Technology, 9014 St. Gallen, Switzerland; tina.buerki@empa.ch; 3Institute of Translational Pharmacology, IFT National Research Council, 90146 Palermo, Italy; stefania.lagrutta@ift.cnr.it; 4Pediatric Clinic, Department of Clinical, Surgical, Diagnostic and Pediatric Sciences, Fondazione IRCCS Policlinico San Matteo, University of Pavia, 27100 Pavia, Italy; amelia.licari@unipv.it; 5Department of Pharmacology and Toxicology, Faculty of Pharmacy in Hradec Králové, Charles University, 50005 Hradec Králové, Czech Republic; frantisek.staud@faf.cuni.cz; 6Consorzio per Valutazioni Biologiche e Farmacologiche (CVBF) and European Paediatric Translational Research Infrastructure (EPTRI), 70122 Bari, Italy; pc@eptri.eu

**Keywords:** gene therapy, somatic cell therapy, tissue-engineered medicinal product, CAR T cell, CAR NK cell, hematopoietic stem and progenitor cell, mesenchymal stromal cell

## Abstract

Advanced therapy medicinal products (ATMPs) are medicines for human use based on genes, cells or tissue engineering. After clear successes in adults, the nascent technology now sees increasing pediatric application. For many still untreatable disorders with pre- or perinatal onset, timely intervention is simply indispensable; thus, prenatal and pediatric applications of ATMPs hold great promise for curative treatments. Moreover, for most inherited disorders, early ATMP application may substantially improve efficiency, economy and accessibility compared with application in adults. Vindicating this notion, initial data for cell-based ATMPs show better cell yields, success rates and corrections of disease parameters for younger patients, in addition to reduced overall cell and vector requirements, illustrating that early application may resolve key obstacles to the widespread application of ATMPs for inherited disorders. Here, we provide a selective review of the latest ATMP developments for prenatal, perinatal and pediatric use, with special emphasis on its comparison with ATMPs for adults. Taken together, we provide a perspective on the enormous potential and key framework parameters of clinical prenatal and pediatric ATMP application.

## 1. General Introduction

Advanced therapies are based on innovative uses or the genetic manipulation of cell and tissue materials, and as treatments for human disease, are often without alternatives or are superior to treatments with conventional drugs. Be it for advanced or conventional treatments, due to regulatory, ethical and commercial pressures, numbers of medical products generally lag behind for pediatric compared with adult applications. However, in particular for advanced therapies, the case can be made that this “age gap” in drug development offers more harm than protection for patients, and that earlier-in-life application, from adult to pediatric or even to prenatal, would have tremendous medical, supply and commercial benefits. In this review, we present the corresponding background and arguments for pediatric and more recently conceived advanced prenatal therapies, by first outlining the current status and challenges of advanced therapies in general, before presenting the conceptual and regulatory framework for early interventions, followed by details for informative preclinical and clinical studies. After discussions of technical and non-technical elements and developments that might facilitate the further and more widespread success of early interventions, the article closes with corresponding perspectives and conclusions.

## 2. Current Status and Challenges of Advanced Therapy Medicinal Products (ATMPs) and Rationale for Early Interventions

### 2.1. Defining ATMPs

ATMPs are medicines that have begun to transform our ability to treat injury and disease. They are defined by the European Medicines Agency (EMA) as comprising gene therapy medicinal products (GTMPs), somatic cell therapy medicinal products (CTMP), tissue-engineered products (TEPs) and combined ATMPs as a combination of any of the three product categories [1]. Specifically for the EU, the EMA provides up-to-date information and relevant guidelines covering their classification [2] and toward marketing authorization [3,4], albeit with a functionally limited search interface. Likewise, clinical trials and new medicines are centrally registered in EMA databases [5,6], but are limited to products and trials approved through the centralized EU procedure, which can be bypassed by application to one or several nation states [7]. Differing terminology and definitions cover ATMPs and their subcategories outside the EU [7,8], which hinders systematic global assessments, but has not stopped an avalanche of comprehensive recent reviews covering their global development, application, regulation, risks and prospects [8,9,10,11,12,13,14,15,16,17,18,19]. For clarity, the present review will refer to advanced therapies in accordance with the most recent quarterly recommendations on the classification of ATMPs by the EMA [20]. Despite this European reference point, and although providing EU-specific references to regulatory procedures and public resources, we will cover findings and prospects for early interventions with a global perspective.

### 2.2. Current Key Areas of ATMP Application

In line with their wide-ranging definition, ATMPs may apply to a multitude of inherited and acquired diseases. However, although preclinical development for ATMPs has already touched on hundreds of conditions, only few major technologies and applications of ATMPs have progressed to the clinical trial stage or received marketing approval. Reasons for this can be found in inappropriate study designs (in terms of numbers, experimental groups, endpoints and comprehensiveness), limited transparency and comparability across studies, high cost and the slow political, regulatory and industry adoption of ATMPs [21]. Among a global pipeline of dozens of approved ATMPs and hundreds of ATMPs in development [22,23], the most advanced ATMPs with proven potential for early therapeutic intervention include receptor-engineered and chimeric antigen receptor (CAR) cells, ex vivo genetically modified hematopoietic stem and progenitor cells (HSPCs) and mesenchymal stromal cells (MSCs) [14]. Of these, CAR cells were initially conceived and developed over several generations of increasingly effective and durable CAR T cells [24], but are now also being developed as CAR natural killer (NK) cells [25] as powerful anti-cancer immunotherapeutic agents for autologous and allogeneic application, respectively [26]. HSPC-based ATMPs have enabled pioneering clinical autologous applications of gene therapies for both gene addition [16,27,28,29,30,31] and gene editing [32], whereas MSCs have also seen widespread use for allogeneic application in different tissues and in the modulation of immune responses (e.g., for Obnitix^®^ and Alosifel^®^) [33,34]. Based on their diverse application and due to their exemplary level of development, as well as for prenatal and pediatric use, CAR cells, HSPCs and MSCs will thus serve as the main examples for ATMP progress and applications throughout this review. Importantly, many additional ATMPs may have great potential for early-in-life applications, but cannot be detailed here. For instance, the diverse category of TEPs, which are applied as autologous or allogeneic cell-based engineered tissues for the regeneration of skin, cartilage and bone, has great clinical significance [35]. However, its successful application in pediatric or prenatal settings is thus far limited to single case studies, safety assessments or small grafts; thus, systematic coverage of TEPs beyond selected landmark examples is outside the scope of this article.

### 2.3. Exemplary ATMP Successes

Among ATMPs under EU regulation, CAR-, HSPC- and MSC-based therapies may offer new prospects to improve the treatment of several conditions with unmet medical needs. For instance, CAR technology has been a trailblazer for processes and regulations governing personalized ATMPs and employs synthetic receptors to direct autologous immune cells to any cellular target without HLA restriction [36]. Current second-generation CAR T cells carry engineered receptors that encode the recognition domain for a tumor-specific epitope, bound via an optional scaffold domain to activation and costimulatory domains, which together facilitate target recognition, cytotoxicity and T cell expansion. The showcase application for CAR technology is autologous CAR T cells against the abundantly expressed and non-essential CD19 B cell marker which, after early success in clinical application to relapsed/refractory B cell malignancies [37], is the basis of currently approved CAR-based treatments (Tisagenlecleucel/Kymriah^®^ EMA/FDA, axicabtagene ciloleucel/Yescarta^®^ EMA/FDA, brexucabtagene autoleucel/Tecartus^®^ EMA/FDA, lisocabtagene maraleucel/Breyanzif^®^ EMA authorization pending/FDA) [24]. With ongoing attempts to target additional markers of malignancies and to even apply CAR technology to solid tumors, the most prevalent targets in addition to CD19 are the less abundantly expressed CD22 and CD20 for B cell malignancies [38,39] and the B cell maturation antigen (BCMA; by idecabtagene vicleucel/Abecma^®^) for multiple myeloma [40]. As additional developments for CAR T cells application, receptor components may be swapped in a modular fashion, site-specific delivery of CARs to the *TRAC* locus in T cells using clustered regularly interspaced short palindromic repeats (CRISPR) technology improves their potency and consistency of expression, and antigen escape by tumors may be reduced or the efficiency of therapy may be enhanced by the application of multiple CARs [41,42] or by bispecific CAR designs [43,44,45]. Finally, employing natural killer (NK) cells instead of T cells confers an improved safety profile by the avoidance of cytokine release syndrome and action on non-hematopoietic cells, and paves the way toward allogeneic (off-the-shelf) CAR cell applications because of the HLA-independent action of NK cells [46].

For HSPCs, autologous transplantation after genetic modification has emerged as the most popular and successful application of gene therapy as a viable option for a variety of monogenic disorders, possibly because of the long experience with allogeneic hematopoietic stem cell transplantation (HSCT) and the ability to manipulate those cells ex vivo with great efficiency [47]. For all these disorders, allogeneic HSCT remains the clinical standard for cure, but is limited by donor availability and associated severe immunologic complications (graft-versus-host disease (GvHD) and graft rejection) [48]. Genetically modified autologous HSPC products, mostly applied in the treatment of blood or immune system disorders, but also of several storage and metabolic disorders, represent an alternative and potentially safer one-off treatment option [49,50]. Despite continuous improvements in the manufacturing of modified cells (refined cell collection, isolation and culture protocols, better vector designs, safer tools based on gene/base editing), and an increasing number of products entering clinical trials for an expanding list of diseases, 30 years after the first ever gene therapy trial, only four HSPC-based GTMPs have been granted EU marketing approval: betibeglogene autotemcel (Zynteglo) for transfusion-dependent beta-thalassemia (TDBT); Strimvelis for severe combined immunodeficiency due to adenosine deaminase deficiency (ADA-SCID); atidarsagene autotemcel (Libmeldy) for metachromatic leukodystrophy (MLD); and elivaldogene autotemcel (Skysona) for cerebral adrenoleukodystrophy (CALD) [51,52,53,54]. All of the above therapies are based on ex vivo viral-vector-mediated gene transfer in HSPCs and the autologous transplantation of modified cells into patients. For in vivo applications of GTMPs, onasemnogene abeparvovec-xioi (Zolgensma) for spinal muscular atrophy (SMA), resamirigene bilparvovec (AT132) for X-linked myotubular myopathy (XLMTM) and voretigene neparvovec (Loxturna) for Leber congenital amaurosis (LCA), are EU-approved therapies based on the systemic or direct/topical delivery of viral vectors encoding functional copies of the respective disease-causing genes [55,56,57]. The above commercially available drugs are indicated either in pediatric-only or mixed pediatric/adult populations, because most of them concern early-onset severe or early-lethal inherited disorders.

Finally, MSCs have become the most clinically studied experimental cell therapy platform worldwide since their first evaluation in humans in 1995 [58]. Although initial insights into MSC properties and mechanisms of action were mainly gained from preclinical murine models and in vitro analyses of human MSCs, their application has already shown enormous potential in the moderation of immune and inflammatory reactions, bone diseases, cancer, and heart, liver or kidney failure, in part through paracrine and exosomal signaling. As exemplary fields of MSC application for immune and inflammatory reactions, acute GvHD (aGvHD) may lead to severe inflammatory reactions and the death of patients after allogeneic HSCT [59], and Crohn’s disease may lead to debilitating chronic inflammation of the gastrointestinal tract [34]. Likewise, acute or chronic injury, such as of the lungs, may irreversibly damage tissue by inappropriate immune responses and/or aberrant repair processes, usually leading to fibrosis and subsequent decline in organ function. MSC attenuation of inflammation prevents further injury and promotes repair [60,61,62], with frequent achievement of full aGvHD remission [63,64], improved bacterial clearance, potential differentiation of MSCs to replace damaged cells, and in part the cytokine-mediated, anti-inflammatory and pro-regenerative action of MSCs [65,66], all vindicating their therapeutic use. Regarding bone diseases, MSC-based therapies act by paracrine- and exosome-mediated signaling, in osteogenesis imperfecta in particular through action of exosomal RNA cargo [67,68], and in bone tissue engineering, scaffolds combined with osteogenically differentiated MSCs provide a better bone regeneration microenvironment and better bone growth than engineered cell-free scaffolds [69,70,71]. For cancer treatment, MSCs may show microenvironment-dependent pro- or anti-tumorigenic properties when unmodified, but clearly assume therapeutic properties when used for the exosomal delivery of cytotoxic agents after drug loading or after genetic manipulation [72,73]. In the treatment of heart, liver or kidney failure, MSCs variably mediate the stimulation of endothelial cells [74], inhibit apoptosis, inflammation and hepatic stellate cell activation [75,76], and deliver trophic factors and cell components by paracrine or exosome signaling and even cell fusion [77], respectively, in order to achieve therapeutic action.

### 2.4. Challenges of ATMP Application

Despite such prominent successes and progress, there are inherent limitations for the widespread use of ATMPs. Many factors determining the cost of ATMP development are those in common with the development of other drugs and treatments, whereas some key cost factors are unique to ATMPs. As for conventional drugs, observations for ATMPs and their efficacy under optimized laboratory conditions do not always reflect the complex and multifactorial reality in the clinic. Moreover, the complex nature of ATMPs and the direct link of their production with live cells from donors and/or recipients suggests that ATMP development is characterized by key challenges not encountered, or of different qualities from those faced, in the development of small-molecule drugs. The great diversity of ATMPs and their applications means that defining universal challenges is difficult; however, several key challenges applying to development, ethical and regulatory issues, supply and manufacturing, and marketing across many different ATMPs are outlined in Figure 1 and detailed in the current section.

#### 2.4.1. Scientific and Medical

On the scientific or medical side, donor-specific features, incompleteness of our understanding of specific mechanisms of interactions with host tissues, and elusiveness of robust pharmacodynamic and pharmacokinetic models for different clinical applications represent major challenges that need to be overcome for each individual disorder to achieve successful clinical translation [49,78]. Especially for in vivo applications, the specificity and efficiency of target cell manipulation are problematic, whereas for ex vivo application based on HSCs, treatment-related morbidity related to myeloablation is a concern. For all ATMPs, addressing prenatal or early-onset diseases in a timely fashion and dealing with immune rejection of treatment and with pre-existing co-morbidities are common challenges.

#### 2.4.2. Ethics and Regulatory

The translation of ATMPs from R&D to clinical trials and then commercialization faces unique ethical and regulatory challenges not encountered at all or to the same degree for small-molecule drugs. Ethically, the inherent complexity of ATMPs and the lack of precedents for many clinical studies are sources of uncertainty, and thus concern, for the safety of participants, whereas ethical considerations for trial participants also routinely prompt trial designs without control groups for what are often invasive procedures. Where inadvertent germline manipulation may be a concern in several in vivo applications, in particular of GTMPs, rare but high-profile illegal human germline manipulation has raised fundamental ethical concerns and have led to a high level of vigilance and stringent regulatory requirements, particularly for GTMP applications [79,80]. Moreover, regulatory requirements for ATMPs are generally high, and are moreover mutable for a nascent industry. Accordingly, ongoing evolution of regulatory, legal, quality-control and infrastructural aspects for ATMPs brings about that framework requirements across different legislatures differ greatly [8,81,82,83]; thus, regulatory challenges are seen as a key difficulty by manufacturers [84].

#### 2.4.3. Supply

In addition to regulatory issues, access to bio- and good manufacturing practice (GMP) materials are a key impediment for ATMP applications. For cell-based therapies, suitable cells are not available in sufficient numbers for corresponding treatments, be it for autologous applications to a single patient or for allogeneic therapies, where the manufacturing of off-the-shelf ATMPs may aim to serve many patients. For certain disorders, the availability of suitable adult stem cells for autologous application presents a challenge even for abundant cells, such as HSPCs or MSPs, in particular where the disorders or injuries to be treated affect the abundance and viability of those stem cells, as is the case, e.g., for HSPCs in Fanconi’s anemia [16], where this may be addressed by improved mobilization and collection [85]. For allogeneic therapies, and even where culture protocols can be established that maintain the desired properties of the cells in question, bottlenecks may be addressed by advanced and scalable culture methods, although high single-dose requirements, e.g., for MSCs of up to 10^9^ cells, or the rarity of starting material may continue to pose challenges for ATMP manufacturing for the foreseeable future [48]. The shortfall thus calls for improved identification, isolation, expansion and modification protocols for autologous application, and for scalable expansion technologies for off-the-shelf ATMPs [48], as is being argued and explored, e.g., for both HSPCs and MSCs in their countless applications [16,49,86,87]. Similarly, the pricing and availability of GMP reagents pose a problem for the clinical development of ATMPs, in that even well-funded clinical trials are held up by limiting global production capacities for GMP-grade reagents. The supply of GMP-grade materials was a limiting factor even before the general pharmaceutical supply chain issues brought about by the COVID-19 pandemic [88].

For many of the challenges of ATMP application, incremental improvements are being made; however, some key problems, particularly the limiting supply of GMP-level therapeutics at population scale, will require a landmark shift in the underlying technologies or in the way they are applied. Even a moderate increase in contemporary GMP requirements by expanding adult application of ATMPs would hardly be sustainable with current technologies [89].

#### 2.4.4. Manufacturing

Supply shortages are aggravated by a lack of standardization of manufacturing and quality control procedures, which still pervades the nascent field of ATMPs, affecting the cost and predictability of production. More specifically for GTMPs, the large size and negative charge of nucleic acids and their consequentially poor penetration of cell membranes calls for the temporary removal of membrane integrity or for dedicated carriers to allow effective delivery to cells and tissues. The development of non-lethal systems for cell penetration and of effective vectors for transfer across membranes has been a long journey from initial attempts at plasmid transfection over advanced non-viral and viral delivery methods to a great diversity of permanent, transient and highly transient delivery of transgenes, of genome editing tools, and of RNA, nuclear reprogramming and epigenome editing reagents today, as we will point out for key developments later.

ATMPs based on autologous cell material are as variable as the patient population, are inherently personalized products that can only benefit from economies of scale in some aspects, and the place and time of manufacture have to be adjusted according to the patients. The corresponding personalized application of ATMPs also contributes to difficulties in trial design and preclinical evaluation as key scientific challenges. However, even for universal products and allogeneic application, variability, sensitivity, quality control and subsequent shelf life are issues of concern for ATMPs far more than for standard pharmaceuticals. Finally, the accessibility of therapeutic tools as targets for the patient’s immune system, and off-target delivery to cells and tissues, where those tools might be unproductive or even toxic, both represent a health and safety risk to patients, but also elevate GMP reagent requirements as an important bottleneck for the industry (see below). Altogether, manufacturing, quality standards and starting materials are therefore seen as key technical challenges for ATMPs [84].

#### 2.4.5. Market and Pricing

On the financial side, funding and reimbursement are key challenges for ATMPs [84]. Be it the long development time combined with fast technology turnover, or be it the high inherent cost of manufacturing ATMPs combined with the uncertainty of manufacturing scale and reimbursement policies, the challenges for commercial ATMP development are enormous [12,90]. Most conventional drugs rely on the chronic application and correspondingly long-term reimbursement of manufacturers for development cost; however, most ATMPs are applied and paid as one-off treatments. The novelty of many ATMPs also means that both physicians and patients often need to be made aware of their existence and benefits, and that their integration into clinical routine, including post-treatment follow-up, often still needs to be established. Finally, just as regulatory requirements differ across legislatures, so do reimbursement models and agreements for ATMPs, which, in turn, creates uncertainty and causes hesitancy or absence of investment for production and commercialization. Therefore, the step from proof of principle to market is particularly difficult for ATMPs, and even for safe and efficient products reaching marketing authorization, the inability of patients or health systems to pay for advanced therapies might ultimately lead to commercial failure and market withdrawal [91].

### 2.5. The Rationale for Early Intervention

A multitude of shortcomings thus bring about that ATMPs currently cannot unfold their full potential while promising a new era of effective cancer treatments, potentially curative treatments for inherited diseases, and off-the-shelf regenerative medicines, among others. It turns out, however, that some of the key impediments to wider and effective ATMP application, including high reagent requirements, ineffective systemic delivery, poor accessibility of target tissues and pre-existing immunity, can be addressed by early intervention, i.e., by application at the prenatal stage, in infancy, in early to late childhood or in adolescence.

Many of the aforementioned challenges are brought about or exacerbated by the restriction of treatment to adults, which, for pioneering treatments and especially clinical trials, is a matter of course, unless the disease in question causes death or irreparable damage in early life stages. The resulting tradeoff for therapy development is that the underlying bioethical guidelines protect the unborn or young life, while at the same time often hiding the true potential of ATMPs by application in adults, where early-onset diseases may already have caused irreparable direct or pleiotropic damage, where affected tissues or therapeutically relevant cells are hard to access or rare, or where requirements for cell material (such as TEPs or stem cells) and GMP reagents are orders of magnitude above what may be required to achieve similar benefits in utero or in pediatric patients. A further argument for the prenatal or pediatric application of ATMPs is the innate proclivity for healing in younger patients and the generally superior regenerative performance of TEPs or stem cells earlier in life. Late treatments are thus limited in their effectiveness due to pre-existing damage and reduced graft performance, exacerbating the universal challenges of target-specific in vivo delivery and stem cell retrieval, and inflating the cost (see Section 6 and Section 7), which thus effectively limits the accessibility of ATMPs to those in need. In utero and pediatric therapies effectively improve all of these critical treatment parameters and may be instrumental in giving more patients and diseases the benefit of advanced therapies. Prenatal treatment, as compared with any postnatal treatment, brings various advantages, including a less developed and thus more tolerant and even tolerogenic immune system [92], higher accessibility of target organs, and reduced target tissue size and cell numbers, with correspondingly lower reagent requirements [93]. However, at a stage where early and in utero ATMP interventions, in particular, are themselves still in their infancy, researchers and clinicians are faced with a dilemma: the earlier the intervention, the greater the potential benefits, but also the greater the potential risks and uncertainties of ATMP applications (Figure 2).

## 3. Conceptual and Regulatory Framework for Early Interventions

Early interventions, although still universally under-researched compared with therapeutic applications in adults, may have an essential role to play in the enhancement of ATMP accessibility and effectiveness. Both pediatric and in utero treatments come with distinct drawbacks and benefits in their practical application, which are subject to resolution and enhancement by ongoing research and development efforts in the laboratory and clinic (see Section 4 and Section 5). Early interventions are correspondingly couched within a conceptual and regulatory framework that, in many aspects, differs quantitatively or even qualitatively from that applied for adult interventions. To avoid the erosion of ethical constraints for early intervention and to warrant benefits and prevent potential harm for patients, establishment, awareness and observance of that framework is essential.

### 3.1. Pediatric

In their pediatric application, ATMPs encounter the same routes of application and problems faced during adult application, but benefit from incremental efficacy, accessibility and cost benefits, and from decremental pre-existing infections and morbidities with lower age and body size, with additional pros and cons, as shown in Figure 2. On the regulatory side, pediatric medicines in the EU have assumed a special role since 2007, when the EU Paediatric Regulation came into force [94], comprising Regulations (EC) No 1901/2006 and its amendment No 1902/2006 [95,96] and aiming to achieve better product information and more pediatric medicines and pediatric research [6]. As a result, in its ten-year report on the implementation of the Paediatric Regulation, the EMA reported an increase in many therapeutic areas in pediatric medicines, albeit with little development for exclusively pediatric diseases or for diseases with distinct pediatric presentation [97]. Likewise, there was an only moderate increase in pediatric research in 2017, from 9.3% pediatric as a proportion of all trials in 2006, to 12.4% in 2016 [97,98]. Both the Paediatric Regulation and the ten-year report acknowledge the dilemma that protecting children from clinical trials for ethical reasons, and as a group that requires legal and regulatory protection, has resulted in a scarcity of specifically pediatric medicines, which has led to the widespread off-label use of adult medicines for pediatric patients, with inherent risks and uncertainties concerning optimal dosage, suitable modes of administration and age-specific side-effects. In recognition of this shortfall of pediatric treatments, the Paediatric Regulation therefore offers incentives for pediatric medicine development to compensate for the cost disadvantage associated with the requirement to investigate age-stratified action for inherently smaller patient populations than would be encountered for adult treatments. These incentives include six-month extensions on the supplementary protection certificate of medicines, and 12 instead of 10 years of market exclusivity for orphan medicines for compliant companies, supported by an advantageous pediatric-use marketing authorization (PUMA) and a specific pediatric expert committee and free advice to the industry from the EMA. This is paired with comprehensive measures and resources for the dissemination of pediatric studies, including an EU network of pediatric investigators, an inventory of pediatric needs, a public database of pediatric studies and the obligation for companies to submit any pediatric data for approved medicines to the regulators for analysis. In line with this and further promoting the dissemination of pediatric trial results, clinical trials for pediatric interventions in the EU are covered in their entirety from phase I to IV in the EU Clinical Trials Register, in contrast to adult interventions, for which phase I trials are excluded [6].

### 3.2. In Utero

In utero therapy offers the possibility of disease treatment and prevention before birth and is considered most appropriate after at least 7 weeks of gestation, when the primordial germ cells are relatively protected from inadvertent germline modifications [99,100]. It can be achieved either transplacentally, by introducing the medication into the maternal circulation, or by direct injection into fetal tissues or circulation. Both approaches call for procedures and reagents that are distinct from those applied in pediatric and adult patients. For transplacental in utero application, the placenta represents a key challenge, as both a route and an obstacle to transfer. A short-lived but constantly developing organ, the placenta separates the maternal and fetal circulations and, at the same time, enables communication between both entities. During approximately nine months of pregnancy, it serves many functions to support fetal development, including the transport of nutrients and gases, immune defense, endocrine signaling for hormone and transmitter homeostasis, and as barrier to protect the fetus against toxins from the maternal circulation [101]. Importantly, and while conventional pharmaceuticals of high liposolubility and low molecular weight traverse the placenta relatively easily, the placenta presents a substantial mechanical and functional barrier and possibly a toxicity target for therapeutics based on genes, cells, or tissue engineering. Therefore, alternatives to transplacental routes of administration are being explored, such as intravenous injection, intra-amniotic administration, or targeted drug delivery approaches [102]. One of the best-researched means of prenatal ATMP administration is ultrasound-assisted injection in the fetal umbilical vein, which allows injected gene constructs or stem cells to bypass the lungs, via the ductus arteriosus and foramen ovale, and to directly enter the fetal systemic circulation. Such fetal injection even offers significant advantages over the early postnatal intravenous administration of ATMPs into a peripheral vein, which often leads to the trapping of cells or gene carriers in the pulmonary microcirculation [103] and reduced systemic exposure. However, direct injection poses the challenge of accessibility and visualization of target tissues and of a developmentally changing anatomy, and the risk of treatment-related injury to mother and child, so that special surgical skills and equipment are required for the procedure [104,105,106]. Together with the unique safety and ethical considerations inherent to the in utero manipulation of human life, these technical requirements currently represent a substantial roadblock for the widespread application of in utero therapies, with additional pros and cons, as shown in Figure 2. On the regulatory side, EU documents currently refer to ATMPs in connection with in utero transfer mostly in the context of inadvertent germline transmission [107]. Importantly, however, the recent EC Guidelines on Good Clinical Practice specific to ATMPs explicitly acknowledge the possible necessity of in utero intervention in severe early-onset conditions or where only early treatment offers benefit, and in general terms recommend the adoption of additional safeguards appropriate for the product, disease and developmental stage [108]. Meanwhile in utero treatments are already applied using conventional drugs, such as for fetal arrhythmias by digoxin or sotalol [109] or for the prevention of fetal viral infection by antiretrovirals in HIV-positive women [110]. Based on growing experience for in utero non-ATMP treatments and from preclinical ATMP studies (see Section 4), it can therefore be hoped that the clinical translation of in utero ATMP applications is not too far in the future [93].

## 4. Preclinical Studies of Early Interventions in Animal Models Using ATMPs

### 4.1. Pediatric

The pharmacological profile of a new medicinal product is generally assessed in adults prior to testing in the pediatric patient population, and efficacy and safety data are extrapolated to children [111]. However, substantial differences in the disease pathology and pharmacological properties of many ATMPs often exist between adults and children, thus necessitating studies in juvenile animal models prior to pediatric application. Alas, dedicated pediatric studies in animal models are few and far between, also because the juvenile time window for mice (the most versatile and most widely used model organism) is fairly narrow, with mice reaching sexual maturity from as early as 23 days after birth for females and usually from 6 weeks for males [112]. As highlighted in a Special Issue assessing pigs as model animals [113], corresponding research therefore greatly benefits from alternative animal models with an extended juvenile period, such as dogs, pigs, sheep and non-human primates [105,113,114,115,116,117,118], which may moreover serve as a large animal model for in utero applications. Importantly, many murine studies of early-onset diseases, although stretching into adult application, are representative of early interventions and therefore also informative in the present context.

Outside our focus on CAR-, HSPC- and MSC-based ATMPs, early treatments in a cornucopia of rodent and large animal models have already been highly informative for a range of ATMPs, target tissues, diseases and juvenile stages, as may be exemplified here, with treatments based on adeno-associated-virus (AAV)-mediated in vivo gene addition. Recent examples include applications in 2–4-month-old sheep to treat Tay–Sachs disease [118] and in early postnatal and juvenile mice to treat CLN3 Batten disease [119], as two exemplary lysosomal storage disorder, as well as in 12-week-old dogs to treat X-linked retinitis pigmentosa [120], and in early postnatal mice, which mimic the human fetal inner ear, to treat congenital hearing loss and vestibular dysfunction in over a dozen studies [121].

For CAR cell applications, the predominance of CAR-T cells and their autologous application largely shifts analyses to toxicity-only assessments in pure animal models, or to more comprehensive functional assessments in murine xenograft models. For the latter, the age of human cell donors is usually not disclosed and recipient NSG mice are usually adult; thus, corresponding publications to date do not allow conclusions drawn from comparisons of the relative performance of early life and adult interventions. Importantly, the contemporary predominant application of CAR cells is cancer treatment by autologous CAR T cells, the performance and developmental age of which are bound up with the age and state of the affected patient. However, in particular for the nascent allogenic application of, e.g., CAR-NK cells, comparative analyses for performance of adult vs. juvenile cells in xenograft models would be extremely informative, because the emergence of standardized protocols and impressive efficacy data indicate for the latest CAR-treatment-based strategies [122]. For instance, the evaluation of second- and third-generation anti-*CAIX* CAR-T cells in NSG-SGM3 mice transplanted with CAIX-expressing clear-cell renal cell carcinoma skrc-59 cells achieved complete remission and tumor-free survival on day 72 after treatment for the best combination of CD4/CD8 CAR-T cell ratio and CAR-T receptor [123]. Evaluation of cord-blood-derived fourth-generation anti-CD19 CAR-NK cells engineered for enhanced cytokine signal transduction achieved the virtually complete suppression of tumor growth in Raji (Burkitt-lymphoma-cell)-transplanted NSG mice, allowing up to 341 days of tumor-free survival, against the death of all control animals in under 1 month [124]. Finally, in the first application of Vδ1 γδ CAR-T cells, targeting GPC-3 as a frequent and abundant marker of solid tumors slowed tumor growth after transplantation of HepG2 cells into NSG mice, down to below 10% of growth in controls, for a test period of over 30 days [125]. This was achieved in the absence of GvHD and toxicity symptoms, because γδ CAR-T, similarly CAR-NK cells, acts HLA-independently and could thus be used as off-the-shelf ATMP.

For HSPCs, murine xenograft models have been crucial for the development and clinical translation of autologous and allogeneic HSPC-based therapies, including gene therapy, since the late 1990s [126]. Immunocompromised mice, humanized mice and genetically engineered disease mouse models have been used to address key bottlenecks of ex vivo and in vivo application of HSPC-based ATPMs, including the maintenance of HSPC multilineage potential, efficient engraftment of cells and safety [127]. Neonatal and juvenile mouse models, in particular, have been extremely useful in the development of HSPC-based gene therapy for primary immunodeficiencies, where early intervention is a sine qua non. For this, the adenosine deaminase deficiency (ADA)^-/-^ neonatal mouse model was used in preclinical studies to assess efficiency and safety of ADA-carrying lentiviral vectors by ex vivo transduction and the autologous transplantation of modified HSPCs, efforts that ultimately led to the EU marketing authorization of Strimvelis in 2016 [128,129,130]. HSPC-based ATMPs have also been tested in neonatal mouse models of further disorders, such as Wiskott–Aldrich syndrome, mucopolysaccharidoses and other storage disorders [131,132,133]. An exceptional achievement is the humanized transgenic thalassemia mouse models developed by the Ryan group which, in contrast, to knockout models, develop transfusion-dependent thalassemia major, and thus are the most faithful representation of the disease and of therapeutic efficiencies in vivo, from in utero to adult applications [134,135,136,137,138]. Larger animal models have also been applied in cases where mouse knockout models have failed to faithfully recapitulate the disease phenotype or where longer-term follow-up has been essential [139]. Among others, juvenile canine leukocyte adhesion deficiency and X-linked severe combined immunodeficiency (X-SCID) dog models, as well as juvenile nonhuman primate X-SCID and human immunodeficiency virus (HIV) models, have been crucial in advancing HSPC gene therapy based on both gene-transferring viral vectors and newer gene editing tools [114,117].

For MSCs, many preclinical trials have been completed and many more are currently ongoing to explore their safety and efficacy in a wide range of acute and chronic disease models, complemented by studies merely investigating the (non-ATMP) application of MSC-derived exosomes and extracellular vesicles. A fundamental limitation for purely murine models (as opposed to xenograft models) in ATMP research is the profound differences for key aspects of MSC biology in ATMP development between murine and human MSCs and the correspondingly poor representation of human therapy applications in studies based on mouse MSCs [140]. These differences concern cell expansion behavior, properties after immortalization or cryopreservation, and choices of paracrine signaling molecules [140,141]; moreover, conclusions across multiple studies are exacerbated by differences between alternative models for the same disease [142]. Therefore, with the swift elimination of human MSCs in immunocompetent mice, xenograft studies based on immunodeficient mice or studies in species for which MSCs or the relevant anatomy more closely reflect human biology [143,144] appear to be the most informative. Based on adolescent to young adult (6–8-week-old) mice, the latest findings for immune and inflammatory disorders include the identification of therapeutic antioxidant and pro-angiogenic action of placenta-derived human MSCs in a surgical model for Crohn’s-like enterocutaneous fistula [145]. Another recent study demonstrated enhanced colonic homing and the enhanced induction of macrophage IL-10 release, through transient CXCR2-receptor and semaphorin-7A expression on human MSCs, respectively, in an immunocompetent murine chemically induced inflammatory bowel disease model [146]. For recent MSC use in GvHD, culture-expanded, high-dose human umbilical-cord-blood-derived (UCB) MSCs were simultaneously transplanted with same-donor UCB HSPCs into NSG mice to effectively suppress GvHD and achieve 60 days of event-free survival [147]. As a particularly striking endorsement of MSCs as therapeutic agents for injury and inflammation, a recent systematic review on preclinical studies testing cell-based therapies in experimental neonatal lung injury, mainly applied in a hyperoxic rodent model of bronchopulmonary dysplasia (BPD), identified MSCs from among 15 distinct cell-derived therapies as the most effective cell-based therapy for key outcomes [148]. BPD in hyperoxia models has been studied for therapies based on MSCs [149]. When delivered intravenously, intraperitoneally or intratracheally, MSCs attenuated neonatal lung injury by decreasing lung inflammatory mediators, such as IL-6 and TNF-a, and reducing the expression of angiotensin II, angiotensin II type 1 receptor, and angiotensin-converting enzyme [149]. MSCs also improved alveolar structure and angiogenesis, inhibited lung fibrosis, and improved exercise capacity in animal models of BPD [150,151,152,153]. Similarly to BPD, acute respiratory distress syndrome (ARDS), which represents a global medical concern with significant morbidity, might also be ameliorated by MSC-based therapies [154]. Experimental in vivo models of lung injury, including acute lung injury (ALI) and ARDS, demonstrated the therapeutic efficacy of MSCs [155,156] or their exosomes, vindicating the corresponding application of MSCs in clinical trials.

Finally, and across different cell and vector systems, several studies in animal models have addressed age comparisons for the efficiency of candidate ATMPs [157,158,159,160,161,162]. For instance, direct AAV-mediated delivery of the *CLN2* gene for treatment of the lysosomal storage disorder, late infantile neuronal ceroid lipofuscinosis, showed clear efficiency advantages for pre-symptomatic compared with post-symptomatic application [160], and a clear survival advantage for younger recipients, with effectively doubled survival time from 2-day- to 3-week- and 7-week-old recipients [157]. For treatment of spinal muscular atrophy, intravenous injection of an SMN-encoding AAV in neonate (postnatal day 1) diseased mice rescued neuromuscular phenotype and life span, in contrast to treatment of 10-day-old mice, followed up by corresponding injection and motoneuron transduction in a neonate wildtype cynomolgus macaque [159]. For leukodystrophy Canavan disease, intravenous AAV injection to deliver the *AspA* gene on postnatal days 0, 6, 13 and 20 retained significant therapeutic action up to day 20, but for all parameters tested gave increasingly better restoration of normal performance with decreasing age at treatment [158]. Finally, young adult vs. aged rats were recently transplanted with bare and allogeneic-MSC-coated vascular grafts to demonstrate superior performance (e.g., for graft integration, blood flow, neotissue density, collagen fiber density and orientation) in young vs. aged recipients and for MSC-coated vs. bare grafts [162], which gives clear implications for MSC inclusion and, despite absence of juvenile animals in the study, once again for a general positive effect of application in younger recipients.

### 4.2. In Utero

Recent years have seen significant progress in the in utero application of different classes of ATMPs. Transfer of stem cells is rarely used, whereas in utero ATMP application by the AAV-mediated direct delivery of gene editing components or therapeutic transgenes is of particular prominence, although direct injection into fetal yolk sac vessels [163,164], ionizable lipid nanoparticles (LNPs) [165] and even in utero electroporation [166] have also been employed to achieve the delivery of cargoes such as mRNA, editors or gene addition components.

Exemplary studies for direct AAV-mediated delivery include early achievements of sustained reporter gene expression in the pulmonary epithelium after injection in the amniotic sac [167] and tolerance induction by the delivery of human factor IX in hemophilia B mice [168]. AAV was also used for the delivery of adenine base editors for efficient correction in the liver and heart and low-level correction in the brain in *Idua^(W392X)^* mutant (Hurler syndrome) mice [169] and that of a β-glucosylceramidase transgene by fetal intracranial injection in *Gba* knockout (Gaucher disease) mice [170] for lysosomal storage disorders. Similarly, CRISPR/Cas9- and cytosine base editors were delivered by intravenous in utero injection to establish proof of principle for in utero editing to modify proprotein convertase subtilisin/kexin type 9 (*PCSK9*) as a target for coronary heart disease and to correct 4-hydroxyphenylpyruvate dioxygenase (Hbd) as therapy for hereditary tyrosinemia type 1 [171]. AAV8-mediated delivery of human factor IX enabled the long-term correction of hemophilia in cynomolgus macaques [105], as did the AAV5- and AAV8-mediated delivery of human factors IV and X [172] in both studies, largely due to randomly integrated provirus in hepatocytes. Finally, AAV2-mediated delivery of MSRB3 by transuterine microinjection into the otic vesicle of MsrB3 knockout embryos has been used to address hearing loss and vestibular dysfunction, which has also been addressed, e.g., by plasmid-based delivery combined with electroporation in connexin 30 knockout embryos [121]. Non-AAV-based direct injection of therapeutic agents has additionally been applied using an adenoviral vector to deliver CRISPR/Cas components and inactivate a mutant *Sftpc^I73T^* gene for partial disease correction in a mouse model of monogenic lung disease [173]. Direct in vivo vector delivery was also performed by the intrahepatic fetal injection of HBB-encoding GLOBE lentiviral vector to achieve the correction of β-thalassemia in a humanized mouse model [137], for comparison with the intraperitoneal delivery of wild-type HSPCs with overall low correction efficiencies [174], and by the intra-amniotic injection of polymeric nanoparticles loaded with triplex-forming peptide nucleic acids and single-stranded donor DNA as gene editing components in *HBB^IVS2−654^* thalassemic mice [175].

Concerning ATMP development based on in utero stem cell transplantation, ex vivo gene addition of human factor VIII to placenta-derived MSCs allowed the detection of postnatal transgene expression after in utero transplantation into wild-type mouse embryos, as proof of principle for a potential corresponding hemophilia A therapy [176]. For HSPCs, in utero application would be based on long-established in utero hematopoietic stem cell transplantation [177] which, due to immunological immaturity [92], can even be employed as conventional fully allogeneic HSCT [177], but in the absence of in utero conditioning regimens, only achieves the mixed chimerism of donor cells for allogeneic or autologous application [177,178]. In this context, amniotic-fluid-derived stem cells show superior performance as a potential substrate for future in utero therapies [179,180]

## 5. Clinical Studies of Early Interventions

### 5.1. Pediatric

A large number of pediatric clinical trials are in progress for ATMPs, and in some cases have already led to approved treatments (Section 2). For TEPs, several studies and applications of tissue engineering for the treatment of skin and soft tissue damage provide notable landmarks with great potential for pediatric application [181,182,183], as was recently demonstrated in clinical studies, e.g., for epidermal autografts (JACE) [184] and composite skin allografts (Apligraf) [183,185]. In the context of genetic skin disorders, two pioneering pediatric studies in seven-year-old boys with junctional epidermolysis bullosa combining retroviral gene addition in autologous keratinocyte stem cells with tissue engineering demonstrated the complete functional regeneration of limited epidermal grafts [186] and permanent clonal reconstitution of the entire epidermis, respectively [187]. These studies at the University of Modena and Reggio Emilia have provided the first concept and then mechanistic insights for wider clinical evaluations of genetically corrected autologous epidermal grafts to treat other genetic disorders, such as Netherton syndrome [188] and recessive dystrophic epidermolysis bullosa [189]. Noteworthy outside our focus on CAR-, HSPC- and MSC-based therapies are also ATMPs with pediatric application developed under the EMA Priority Medicines (PRIME) scheme [190]. Corresponding pediatric treatments still under investigation and pending approval include rebisulfigene etisparvovec (ABO-102), an AAV9-based gene therapy drug for the in vivo treatment of mucopolysaccharidosis IIA-Sanfilippo syndrome, and beremagene geperpavec (KB103), an HSV-1 collagen-expressing vector for the topical treatment of dystrophic epidermolysis bullosa, both of which are currently in clinical trials (NCT04088734, NCT04360265, NCT02716246 and NCT03536143, NCT04491604, respectively). A third pediatric ATMP, AT-GTX-501, based on an AAV9 vector containing the human CLN6 gene to slow disease progression in variant late infantile neuronal ceroid lipofuscinosis 6, has recently been discontinued because it failed to stabilize disease progression in long-term follow-up (NCT04273243 and NCT02725580) [191].

#### 5.1.1. CAR Cells

For CAR cell application, CAR-T cell-based therapy is exceptionally advanced and too prolific to cover here in detail for all corresponding clinical trials, with marketing approval for several ATMPs for pediatric application (see also Section 2). Notable here is the EMA and FDA ACCELERATE collaboration for pediatric cancer patients [192] which, among other ATMPs, led to marketing approval as priority medicines under EMA’s PRIME scheme [190] for the CD19-targeting, CAR-T-based tisagenlecleucel (Kymriah^®^, e.g., NCT02529813 and many others) for pediatric patients with relapsed or refractory B cell acute lymphoblastic leukemia, whereas for other equivalent treatments, such as axicabtagene ciloleucel (Yescarta^®^, e.g., NCT02348216 and many others), data are still pending that would warrant treatment in pediatric patients. Overall, over 60 CAR-T trials for pediatric application are currently open, recruiting or ongoing (see www.clinicaltrials.gov, accessed 21 February 2022), with targets such as acute lymphoblastic leukemia (anti-CD19, anti-CD22, anti-CD19/22, anti-CD20/19), CNS tumors and sarcomas (anti-B7H3, anti-HER2, anti-EGFR806), Hodgkin lymphoma (anti-CD30), liver cancer (anti-GAP), neuroblastoma (anti-GD2), myeloid leukemia (anti-CD33, anti-CD123) and T cell lymphoblastic leukemia (anti-CD7).

#### 5.1.2. HSPCs

For advanced HSPC-based therapies, promising preclinical data have led to dozens of products entering clinical trials during the last few years. Specifically, for early-lethal inherited disorders, clinical trials of related drugs exclusively involve children; otherwise, a mixed adult/pediatric or, more often, an adult-only cohort is used. The juvenile application of HSPC-based ATMPs in children, mostly as gene therapy of inherited disorders, comes with its unique challenges and opportunities [193]. Early application is paramount for many HSPC-based GTMPs, especially when the corresponding disease poses an immediate threat to life or causes early irreversible damage, but also because early drug administration is associated with better outcomes due to better health status, bone marrow (BM) condition and quality of stem cells [194,195]. This benefit of earlier intervention, highlighted in a plethora of preclinical studies in animal models (see Section 4), as well as in clinical studies of allogeneic HSCT [196], has only recently begun to show in gene therapy clinical studies, strongly advocating for improvements in prenatal screening and early diagnosis [197]. Moreover, early treatment reaches therapeutic effects at a lower price (lower body mass requiring lower drug dose), abolishes or reduces long-term medication requirements and prevents disease complications, together making an otherwise expensive and inaccessible treatment rather cost effective [198,199,200].

Two out of the three EU-approved HSPC-based GTMPs, Strimvelis for ADA-SCID and atidarsagene autotemcel (Libmeldy) for MLD, are specifically indicated in young pediatric patients [201,202], whereas betibeglogene autotemcel (Zynteglo) for TDBT is indicated in children >12 years old and adult patients [197]. Lybmeldy in particular was given authorization for use only in children with late infantile or early juvenile MLD who are asymptomatic or have initial symptoms but can still walk independently and do not show mental deterioration, as the drug showed much less benefit in children with a more advanced disease stage [54,98,202]. As yet another advocate for early intervention, a recent gene therapy study for β-thalassemia, employing autologous HSPCs after lentiviral vector-mediated *HBB* gene addition in patients, including pediatric patients, allowed the direct comparison of pediatric and adult treatment with HSPCs and showed that a younger age is associated with better clinical outcomes (NCT02453477). This was attributed to an impaired BM microenvironment (as the recipient tissue of modified stem cells) and HSPC repopulating capacity in older patients, due to both aging and an advanced disease pathology [194,203].

Due to the large number of clinical studies involving HSPCs as GTMPs, Table 1, in addition to selected studies employing MSC, lists only studies involving products with pediatric application that are currently under the EMA PRIME scheme.

#### 5.1.3. MSCs

The success of MSCs in preclinical models has, over the past 10 years, prompted investigation of their regenerative potential for stem-cell-based therapies in the treatment or prevention of GvHD and in different lung diseases of infants and children, such as BPD, pneumonia, ALI and ARDS [204].

*GvHD*. Typical for the pediatric development of new medicines, many products successfully applied in adult patients do not or only with delay find application in pediatric patients. For instance, whereas for Alosifel^®^ (aka Darvadstrocel) no data exist for patient groups from 0 to 17 years [205], Obnitix^®^ has been employed successfully in pediatric patients with steroid-refractory acute graft-vs-host disease (aGvHD) [33]. Despite this relative shortfall of pediatric development, a gratifyingly large selection of clinical studies has nevertheless already shown the benefit of allogeneic MSC treatments for aGvHD correction or prevention in pediatric patients or for the promotion of engraftment [63,206,207,208,209,210,211]. Of note, the source and nature of the MSCs applied affected the outcome, in that the co-infusion of MSCs effectively prevented aGvHD, but umbilical-cord-blood-derived MSCs additionally improved engraftment, in contrast to parental-derived MSCs [212,213].

*BPD*. BPD is a main complication of prematurity, resulting in significant morbidity, mortality and lifelong consequence of early impairment [214], emphasizing the potentially critical role of early intervention. An apparent reduction in the BPD of lung-resident stem/progenitor cells from the endothelial, mesenchymal and epithelial lineages [215,216] as possible cause for lung growth has suggested a potential therapeutic role of MSCs in preterm infants. This prompted a pioneering phase I dose-escalation trial based on treatment with umbilical-cord-blood-derived MSCs in nine preterm infants at high risk of BPD, which was well tolerated without any serious adverse effect (NCT01297205) [217]. A subsequent two-year follow-up study confirmed the lack of any long-term side effect in the treated patients [218], which has now been extended up to 5 years of age (NCT02023788) [219]. A similar phase I dose-escalation trial on the safety and feasibility of intratracheal MSCs in twelve preterm infants at high risk of BPD yielded comparable results (NCT02381366) [220] as encouragement for further trials including controls and a greater sample size.

*ARDS*. ARDS is a life-threatening condition with acute hypoxemic respiratory failure and other cardio-pulmonary features [221], with a range of possible environmental causes and underlying conditions [222] and the contribution of dysregulated inflammatory/immune responses, coagulation, and alveolar membrane permeability in its pathogenesis [223]. Pediatric mortality is high [154] and supportive care is inadequate, which renders ARDS an ideal target for evaluations of pediatric MSC treatment. Three adult-only trials (phases 1, 2A and 2B) have given encouraging results for an intravenous dose of 10^6^ cells/kg (NCT01775774, NCT02097641 and NCT03818854) [224,225,226], with the need to improve MSC viability and to demonstrate safety and efficacy in pediatric patients.

**Table 1 pharmaceutics-14-00793-t001:** Exemplary clinical in utero and pediatric trials of ATMPs.

Cell Type	Target Disease	Drug ^1^	Drug Short Description	NCT ID	IU/P/A	*n*	Ref.
CAR-T cell	Relapsed/Refractory HL	CD30.CAR-T	CD30-directed genetically modified autologous T cells	NCT04268706	P/A	14 (recruiting)	[227,228]
HSPC	TDBT	CTX001	Autologous CRISPR-Cas9 modified ex vivo CD34+ cells	NCT03655678	P/A	15	[229,230]
HSPC	SCD	CTX001	Autologous CRISPR-Cas9 modified ex vivo CD34+ cells	NCT03745287	P/A	7	[229,230]
HSPC	TDBT	OTL-300	Autologous CD34+ cells transduced ex vivo with a lentiviral vector (GLOBE) encoding the *HBB* gene.	NCT02453477 NCT03275051	P/A P/A	9 9	[231] [194,232]
HSPC	LAD-1	RPL-201	Autologous CD34+ cells transduced ex vivo with a lentiviral vector (Chim-CD18-WPRE)encoding the *ITGB2* gene	NCT03812263	P/A	7	[233,234,235]
HSPC	MPS-IH	OTL-203	Autologous CD34+ cells transduced ex vivo with a lentiviral vector (IDUA LV) encoding the IDUA gene.	NCT03488394	P	8	[236,237]
HSPC	XSCID	MB-107	Autologous CD34+ cells transduced ex vivo with a lentiviral vector (CL20-i4-EF1α-hγc-OPT) encoding the *IL2RG* gene.	NCT03315078 NCT01512888	P/A P	5 (recruiting) 8 (recruiting)	[238,239] [240,241]
HSPC	SCD	ECT-001-CB	UM171-expanded cord blood	NCT04594031	P/A	Recruiting	[242]
HSPC	High-Risk Myeloid Malignancies	ECT-001-CB	UM171-expanded cord blood	NCT04990323	P/A	Recruiting	[243]
HSPC	FA	RP-L102	Autologous CD34+ cells transduced ex vivo with a lentiviral vector (PGK-FANCA-WPRE) encoding the *FANCA* gene.	NCT03814408 NCT04248439 NCT04069533 NCT04437771	P P/A P P/A	25 (recruiting) 5 (recruiting) 9	[244,245] [246] [247] [248]
T cell	Serious viral infections in allogeneic HSCT recipients	Posoleucel (ALVR-105)	Allogeneic multi-virus specific T lymphocytes	NCT04693637 NCT04390113	P/A P/A	12 (recruiting) Recruiting	[249,250] [251]
MSC	BPD	Pneumostem^®^	Intratracheal delivery of umbilical cord MSCs, 1–2 × 10^7^ cells/kg BW	NCT01297205 NCT01632475 NCT01828957 NCT01897987 NCT02023788 NCT02381366	P P P P P P	9 9 33 (T) + 33 (C) 62 8 12	[217] [218] [252,253] [254] [219] [220]
MSC	MMC	PMSC-ECM	Placental delivery of PMSC-ECM	NCT04652908 (CuRe)	IU	35 (T) + 20 (C) (recruiting)	[255]
MSC	OI	Boost cells	Intravenous injection of first-trimester-derived allogeneic expanded fetal liver MSCs	NCT03706482 (BOOSTB4)	IU/P	15 (T) + 15 (C) (recruiting)	[256]

^1^ Treatments with in utero or pediatric aspects under clinical investigation and currently supported by the EMA Priority Medicines scheme (access date, 1 February 2022), with the exception of MSC-based studies. A: adult; BPD: bronchopulmonary dysplasia; BW: body weight; C: control arm; CAR: chimeric antigen receptor; CRISPR-Cas9: clustered regularly interspaced short palindromic repeats-Cas9; FA: Fanconi anemia; HL: Hodgkin lymphoma; HSCT: hematopoietic stem cell transplantation; HSPC: hematopoietic stem and progenitor cell; IU: in utero; LAD-1: leukocyte adhesion deficiency type 1; MMC: myelomeningocele (spina bifida); MPS-IH: mucopolysaccharidosis type IH (Hurler syndrome); MSC: mesenchymal stromal cell; *n*: number of participants (+ control/reference-treated patients); OI: osteogenesis imperfecta (brittle bone disease); P: pediatric; PMSC-ECM: placental MSCs seeded on an extracellular matrix; SCD: sickle cell disease; T: treated arm; TDBT: transfusion-dependent beta-thalassemia; XSCID: X-linked severe combined immunodeficiency.

### 5.2. In Utero

Due to uncertainties and bioethical concerns associated with nascent technologies of in utero ATMP application, few prenatal therapies are currently in clinical trials, and those that are aim squarely at preventing or ameliorating severe diseases with in utero onset, rather than reducing costs or increasing therapeutic efficiency in diseases otherwise suitable for postnatal ATMP application. In addition to animal studies for in utero ATMP application (see Section 4), non-ATMP in utero therapies such as curative intra-amniotic administration of ectodysplasin A protein for hypohidrotic ectodermal dysplasia [257] have paved the way for registration of the first in utero ATMP clinical trials.

For HSPC application, *hydrops fetalis* in alpha-thalassemia major represents a textbook case of the need for and unique potential of in utero therapeutic applications. With in utero blood transfusions as a powerful life-saving technology [258], in utero HSPC transplantation was a logical next step in a conventional HSCT trial with planned enrolment of 10 patients (NCT02986698) [259,260], and possibly paving the way for pioneering in utero gene therapy applications in the clinic, based on autologous HSPCs.

Drawing on MSCs, the in utero transplantation of placental MSC is envisaged in a trial addressing myelomeningocele (aka spina bifida; NCT04652908) as supportive treatment for successful outcomes of in utero myelomeningocele surgery, with the planned enrolment of 35 patients for combined treatment and 20 patients as surgery-only controls [255]. Based on allogeneic fetal expanded MSCs, another study addresses osteogenesis imperfecta (NCT03706482) with administration of three doses in utero in the treatment group vs. three doses starting at 4 months after birth for the control group [256].

## 6. Tools for Success of Early Interventions

In addition to benefits concerning therapeutic efficacy specifically for ATMPs, juvenile treatments in general cater for a large and growing market, as detailed elsewhere in this Special Issue [261], and as for any research and medical sector, growing application will lead to the creation of additional resources. In this respect, progress across all aspects of ATMP development will benefit from early interventions, but of particular importance here might be the recent developments for cell sources, vector development, and in particular, the burgeoning field of nanomedicine, as detailed subsequently.

### 6.1. Sources for Cell-Based Therapies

For CAR cells, T and NK cells as the substrate for the generation of CAR-T and CAR-NK cells, respectively, have a variety of abundant sources. For T cells, typically autologous peripheral blood mononuclear cells (PBMCs) are collected by leukapheresis before the isolation of T cells by CD3 selection [262]. T cells then require activation before transduction to create CAR-T cells, and activation is most reproducibly achieved with anti-CD3/anti-CD28 antibody-coated beads. Transduction is then typically performed by lentiviral or γ-retroviral vectors, with the preference of lentiviral vectors due to their safer integration profile [263] and effective transduction of quiescent cells [264]. For NK cells with their HLA-independent action, autologous but also allogeneic cell sources are suitable, including induced pluripotent stem cells, NK cell lines, umbilical cord blood or allogeneic PBMCs [265]. Depending on the cell source, such as PBMCs, irradiation of CAR-NK cells may not be necessary, whereas for other cell sources, such as the commonly used, cytotoxic and highly passaged NK92 cell line, irradiation is required to prevent malignancies [266]. Based on PBMCs, CD3-negative followed by CD56-positive selection is usually employed to isolate NK cells before activation and transduction. Importantly, off-the-shelf NK cell lines for specific malignancies are already available, such as CD38/BCMA-targeting FT576 line for multiple myeloma [267], and the CD19-targeting FT596 line for B cell malignancies [268].

In HSPC-based cellular therapies, BM, mobilized peripheral blood and umbilical cord blood are all popular sources of HSPCs, each with their own set of advantages and disadvantages, determined by differences in collection procedures, cellular content (cell types and numbers) and outcomes of transplantation [269]. The same tissues serve as HSPC sources for ATMP-related pediatric applications; however, children as donors or recipients of HSPCs face their own unique challenges [270,271,272]. For decades, BM has been the gold standard source of HSPCs in children; however, in recent years, an increasing number of centers has instead used mobilized peripheral blood as the primary source of HSPCs [273]. The procedure is less invasive than BM harvesting, and also characterized by the more rapid engraftment of HSPCs after transplantation [274]. For the collection of mobilized peripheral blood, mobilization agents such as G-CSF and plerixafor are administered to the patient to allow the rapid egress of HSPCs from BM into peripheral blood, from where they are then collected by a blood cell separator (apheresis machine) [275]. The procedure is more challenging in children, especially in those under 10 kg (higher risk of hypovolemic shock, hypocalcemia, hypervolemic cardiac overload and adverse events related to insertion of dialysis catheters), requiring more time and money than in adults [272]. There is, therefore, considerable scope for adjustment to pediatric needs, both for current procedure protocols and for existing adult-oriented technology, such as standard apheresis machines with their large extracorporeal volume in relation to the total blood volume of small children [276]. Even though higher HSPC yields are obtained during harvest in children compared with adults, the scarcity of stem cells with long-term repopulating potential (present among large numbers of committed progenitors or mature blood cells) in the available sources requires the enrichment and expansion of those cells in cultures to achieve therapeutic doses of products. Improvement and innovation in every step of source cell processing, from mobilization (newer mobilizing regimens to maximize HSPC harvest) to apheresis (to increase cell yields and minimize associated risks) and cell culture procedures (to retain stemness and long-term repopulating capacity of cells) are needed for the industrial translation of HSPC-based therapies [277,278,279].

MSCs are a heterogeneous subset of multipotent adult stem cells present in multiple tissues of different sources. Human MSCs can easily be isolated from the umbilical cord, BM, and adipose tissue, and when expanded in vitro can differentiate into different mesodermal cell linages with exceptional genomic stability and few ethical issues [280]. These characteristics have marked their importance in cell therapy, regenerative medicine and tissue repair. BM and adipose tissue are well characterized and documented sources of MSCs. When selecting adequate sources, clinicians should consider some practical limitations concerning the difficulty and invasiveness of the procurement process and various donor characteristics [281]. For BM-derived MSCs, harvesting these cells is a painful, invasive procedure, with a risk of viral exposure and with a potential reduction with donor age in the number, differentiation potential and maximal life span of BM-derived MSCs [282]. Instead, a large number of MSCs can be obtained from the adipose tissue through minimally invasive lipoaspiration methods [283], which maintain their potency with increasing donor age and possess a more robust immunomodulatory capability than BM-derived MSCs [284]. Finally, umbilical cord (UC)-derived MSCs exert faster self-renewal properties than BM-derived MSCs and can be obtained with a painless collection procedure from Wharton jelly, veins, arteries, the umbilical cord lining and the subamnion and perivascular regions [285].

Of note, for off-the-shelf application of any cell type, cell material needs to be expanded, which may be enhanced by new developments in advanced expansion technologies. This might also be necessary for autologous applications, where the underlying disease condition affects cell yield and function. A key step here is a transition from yield-limiting planar to multi-layer and target-cell-optimized microcarrier systems, where porous microcarriers may be of particular benefit for MSC expansion [48]. Alternatively, early ATMP application, with its ballpark drop in reagent requirements and cost, may effectively address the challenge of providing sufficient cell material instead.

Once collected and possibly expanded, cells of interest need to reach their application target, a process for which the selection of preclinical and clinical studies in Section 4 and Section 5 indicates a range of different successful cell delivery modalities for early interventions. Accordingly, intravenous injection readily allows CAR-T and -NK cells to find their engineered, receptor-specific targets, and allows HSPCs, MSCs and many other cell types to home to their tissue of origin. Alternatively, the intraosseous application of HSPCs has advantages for the speed of reconstitution [194], and targeting of cells to the CNS (where generally direct vector injection is preferred), the eye or the embryo with their corresponding transport barriers would usually altogether rely on topical delivery instead [286,287,288].

### 6.2. Traditional Viral and Non-Viral Vectors

The plethora of diverse ATMPs requires the use of different delivery vehicles and routes of administration of cells and genetic material for the achievement of optimal therapeutic effects in adults and children.

For the delivery of GTMPs, efficient transfer of genetic material or genome editing tools into target cells is a key step for success. For both ex vivo and in vivo applications, various viral and non-viral vectors have been developed as delivery vehicles, each one with their own advantages and disadvantages [289,290]. Improvements can be achieved by ongoing technical innovations, such as in the delivery of cells or genetic material to the tissues and cells of interest for in vivo application [291,292,293,294,295,296,297] or for ex vivo application in the isolation of suitable cells [16,298,299,300] or in more effective delivery into cells, and possibly nuclei, at improved efficiency and low toxicity [87,301,302,303,304,305,306]. Viral vectors (γ-retroviruses, lentiviruses, adenoviruses and adeno-associated viruses) were among the first exploited delivery platforms and are still highly prevalent due to their inherently high efficiency of gene transduction to eukaryotic cells (extensively reviewed in [106]). Here, AAVs in particular are almost universally exploited for in vivo application, whereas lentiviral vectors are usually the vectors of choice for permanent ex vivo HSPC modifications. Traditionally, ex vivo gene therapy has been applied to HSPCs, which is still associated with chemical myeloablation and corresponding treatment-related morbidities. Recent advances in antibody-mediated conditioning, such as through the targeting of CD117 cells, promise to improve the tolerability of HSPCs modified ex vivo and reduce treatment-related mortality [307]. Meanwhile, non-hematopoietic tissues are usually modified by in vivo strategies, which poses the problem of accessibility or homing to target tissues but has lower treatment-related morbidities. Lately, in vivo therapy is also being pursued for HSPCs as a safer, cheaper, and potentially more efficient therapeutic approach. Hemoglobinopathies, as the most common monogenic disorders, are once more paving the way as a test bed for new methodology, with the recent publication of several in vivo [293,308,309,310] and one in utero application of GTMPs [137]. Viral gene delivery is frequently associated with considerable immunogenicity and risks of genotoxicity, although many non-viral delivery methods have disadvantages of their own, including frequently lower transfer efficiency, and reduced specificity and duration of gene expression [311]. The fast-paced contemporary research field of non-viral vectors covers polymers, lipids, inorganic particles, engineered virus-like particles, hybrid systems of these vector types and naked nucleic acids for chemical or physical transfer [289,310,312]. For GTMPs, gene editing represents a special field of delivery application, because the persistence of editors would be detrimental and permanent changes in target cells can be introduced by highly transient action instead. The latter is most frequently achieved by electroporation ex vivo, whereas in vivo AAV-based delivery currently predominates [313], as a compromise between the desired efficiencies and the disadvantages of long persistence, low payload capacity and concerns over using high titers of viral vectors. Toward clinical application, there is, therefore, an increasing need for in vivo delivery technology with tissue specificity, flexible half-life, high payload capacity, reproducibility, GMP compliance and low immunogenicity. For traditional vectors, these properties are often difficult to achieve.

### 6.3. Nanomedicine

Nanomedicine has the potential to transform the delivery of therapeutic transgenes by providing highly versatile nanoparticle-based delivery platforms with improved safety profiles. Small particles in the nano-size range (at least one dimension < 100 nm [314]) can nowadays be engineered at large scales and with high precision to enable non-personalized as well as precision therapies [315]. Moreover, recent advances in nanoparticle designs to incorporate complex architectures, bio-response moieties and targeting agents allow for substantial control over their interactions with biological environments and help overcome biological barriers [315]. Therefore, nanoparticles can be tailored, among others, to protect the transgenes from degradation by nucleases, to reduce the stimulation of immune responses or to selectively target specific tissues or cell types to allow for maximum efficiency and minimal off-target effects. The genetic payload itself can be either entrapped into the nanoparticles or attached to the particle surface [316,317].

Side effects are of particular concern for early therapies in pregnancy, because the safety of the pregnant mother and the highly vulnerable developing fetus are at stake. In this context, the placenta, at the interface between maternal and fetal tissues, is critical for fetal development [318]. It governs active and passive gas, nutrient, hormone and waste transport, while blocking many larger molecules from passage to the fetus. Importantly, drug characteristics, such as acid/base properties, hydrophobicity or size, may affect selective passage and the potentially harmful accumulation of drugs in fetal, maternal or placental tissues. Nanoparticle-based delivery of therapeutics (e.g., chemical compounds, biologics and nucleic acids) may utilize such effects on placental translocation and interactions in order to facilitate the specific targeting of maternal, placental or fetal tissues for a highly targeted treatment and for the prevention of off-target effects [319]. For instance, the correction of maternal diseases will require a nanoparticle design that does not allow placental tissue accumulation or fetal translocation, whereas therapeutics for placental complications would rely on nanocarriers that preferentially locate to this particular organ. For fetal therapies, nanoparticles can be administered to the maternal circulation if particles with high placental transfer and specific targeting moieties for placental tissues can be identified. Alternatively, they can be injected directly into the amniotic fluid, umbilical vein or specific fetal tissues to bypass the placental barrier. However, even if the fetus is targeted directly, it will be important to ensure minimal fetal to maternal particle transfer. A recent study has proven the potential of in utero fetal gene editing by showing that the intra-amniotic administration of polymeric nanoparticles containing peptide nucleic acids (PNAs) and donor DNAs was able to correct a disease-causing mutation in the β-globin gene in a mouse model of human β-thalassemia [175]. In another study, PLGA nanoparticles were used for efficient delivery of the CRISPR-complex (Cas9 protein, single gRNA and a fluorescent probe) into erythroid cells in vitro to elevate fetal globin expression [320]. Interestingly, the initial burst release of the content was followed by a sustained release pattern, indicating that intelligent nanocarrier designs could be further exploited to control for the release of the payload according to the therapeutic needs, e.g., to achieve fast release for genetic editing versus slow or sustained release for epigenetic or RNA editing.

In the past decade, significant efforts have been made to understand nanoparticle transport across the placenta in dependence of their physico-chemical properties and to identify targeting signals to direct their localization to specific tissues (mostly the placenta) [175]. Particle size is a key factor to affect placental translocation with a negative correlation (higher transfer for smaller particles), but other particle properties, such as surface charge, material composition or surface ligands, have an impact as well. Due to this complexity, it is still difficult to predict the placental transfer of a nanoparticle, and most likely a combination of multiple particle characteristics will determine its transplacental transport behavior. In addition to passive targeting approaches by the modulation of physico-chemical particle properties (e.g., size, charge, hydrophilicity, shape and chemical composition), active targeting strategies can enable the delivery of the payload to specific cell types or tissues. Several research teams have screened for and identified peptides or antibodies to target placental tissue [321,322]. For instance, Li et al. [323] have conjugated peptides targeting chondroitin sulfate A (CSA; expressed at the membrane of placental trophoblasts) to the surface of nanoparticles to deliver siNRF2 and sisFlt-1 to the placenta, which improved maternal and fetal outcomes in a preeclampsia mouse model [323]. Although siRNAs are not considered as ATMPs [324,325], this study highlights the feasibility of nanocarriers for the targeted placental delivery of genetic material to improve placental functions. In fact, proper placental function is essential for successful pregnancy, and consequently, placental dysfunction is involved in the pathogenesis of many pregnancy complications (e.g., intrauterine growth restriction, preeclampsia, preterm birth). In addition, recent work from Singh et al. indicates that persistent DNA damage in the placenta affects embryonic health, which emphasizes the importance of genome integrity for placental health and embryonic development [326]. Therefore, in addition to maternal or fetal therapy, the placenta could be an interesting target for ATMPs to improve health outcomes in complicated pregnancies and to reduce adverse health effects later in life.

Research on nanoformulations of chemical compounds, biologics or nucleic acids for in utero or pediatric use is still in its infancy, but slowly gathering momentum. In general, the main classes of nanoparticles exploited in nanomedicine applications are polymer-based, lipid-based, inorganic or dendrimer nanoparticles [315,327]. Most prenatal nanotherapies employ non-ATMP cargo, such as conventional drugs, siRNA and proteins [327,328], but there is growing interest to apply nanocarriers for gene addition and gene editing in pregnancy. In fact, the first transplacental gene delivery using plasmid DNA:lipopolyamine complexes to achieve non-invasive fetal drug delivery was reported as early as 1995 [329]. More recent examples explored transferrin-targeted PEGylated immunoliposomes to deliver plasmid DNA to fetal brain [330] or internalizing the arginine–glycine–aspartic acid (iRGD)-coated diblock copolymer complexed to hIGF-1 plasmid DNA under the control of trophoblast-specific promoters (Cyp19a or PLAC1) to improve fetal growth restriction [331]. Although there are evidently tremendous opportunities for novel nanoparticle-based ATMPs, there are still some challenges ahead concerning the efficiency, stability and toxicity of nanocarriers [315]. These will need to be addressed comprehensively, such as by new intelligent nanoparticle designs, in order to achieve the full potential of nanoparticle-based delivery platforms and to establish safety for routine clinical application. In parallel, ongoing systematic assessments of patient safety, as well as of occupational and environmental risks along the life cycle of corresponding medicinal products and ATMPs, is required to share the information with all parties involved, including regulators, consultants, manufacturers, physicians and patients [332,333].

Systemic delivery via intravenous infusion is the most common approach for GTMP administration, although direct/local delivery of treatment into affected tissues (e.g., BM and the liver) is also used depending on clinical application [16]. Recently, the direct delivery to the fetal liver, lungs and intestines by the injection of mRNA-loaded LNPs into the fetal vitelline vein has been performed to achieve corresponding protein expression in the fetal liver [165]. For MSCs, both systemic delivery (intravenous/intraarterial infusions) or local/direct delivery (e.g., intramuscular and intratracheal injections) of cells have been tested in several preclinical and clinical studies [334]. In pediatric clinical trials of MSCs for aGvHD and bronchopulmonary dysplasia, intravenous and intratracheal routes are used, respectively [335,336], whereas for the limited in utero applications of MSCs for prenatal treatment of congenital diseases such as osteogenesis imperfecta and myelomeningocele, infusions via the umbilical vein or local/direct intraspinal infusions are administered [67,255].

## 7. Non-Technical Considerations for the Routine Application of Early ATMP Interventions

As technology, preclinical and clinical development has progressed to facilitate early interventions, a clear majority and a large proportion of the public express their approval of pediatric and of in utero applications of gene therapy to treat inherited diseases, respectively [337]. Beyond general attitudes, financial considerations clearly favor a shift from adult to early interventions for safe and efficacious treatments, whereas many ethical and regulatory impediments that remain for the still-developing ATMP sector (see Section 2) are further exacerbated, as detailed subsequently.

### 7.1. Financial Considerations

ATMPs are typically priced between USD 18,950 for tissue-engineered products and USD 1,206,751 for gene therapy, with the aforementioned prices excluding procurement, inventory and administration costs [338]. The high sales price of marketing-approved ATMPs can be attributed to a variety of factors, including cell sourcing (i.e., cell/tissue acquisition and expansion), GMP manufacturing (i.e., labor-, time- and cost-intensive GMP protocols and procedures, costly clinical-grade reagents and stringent quality control), distribution, and clinical application, including treatment and long-term follow-up [339]. The highly personalized nature of ATMPs, which restricts the scalability of manufacturing pipelines, and the small number of patients qualifying for these treatments, further add to the high prices of ATMPs. An examination of concrete cost factors specifically for ATMPs highlights the benefit of early interventions, in particular for the procurement of starting material and GMP manufacturing (including storage and distribution) and quality control (Figure 3).

The extraordinary cost of ATMPs compared with small-molecule drugs is most readily accepted where ATMPs represent potentially curative treatments, allowing financial comparisons of one-off cures vs. lifelong palliative treatments. However, for many ATMPs, there are additional, less tangible financial benefits. Even for ATMPs with the uncertainty of truly curative outcomes, considerations of permanently reduced disease severity for ATMP applications, of potentially catastrophic financial and health consequences for chronic palliative treatments, and of hope for a curative ATMP outcome or for prolonged survival and potential access to improved future therapies, should enter pricing considerations, in particular for ultra-rare diseases [198]. All three aspects strongly favor early intervention.

For the specific case of GTMPs, a major cost is that of vector production for the delivery of genetic materials. Here, juvenile or prenatal application would allow a ballpark change in materials, and thus, cost, per patient, in addition to reducing requirements for what is frequently limiting cell material for ex vivo GTMPs. Assuming an average body weight of 70 kg [340], a neonatal weight of 3.5 kg [341] and approximately 3 × 10^5^ cells for in utero therapeutic intervention [137,342], an assumed vector cost of USD 100,000 per adult patient [89] at, e.g., 5 × 10^8^ lentiviral transduction units per kg [194], would be reduced to USD 5000 in neonates and to USD 85 for in utero treatment [137,194,342]. Before any markup of commercial products and treatment-associated cost, this change in pricing would greatly increase accessibility of treatment.

### 7.2. Ethical and Regulatory Considerations

From the ethical and regulatory standpoint, de novo pediatric developments or adaptations of adult treatments are impaired by the absence of a consensus for the establishment of pediatric safety specifications, as analyzed elsewhere in this Special Issue [343]. Consequently, even for conventional small-molecule drugs, pediatric applications lag far behind developments for adults [344,345], as is also apparent in the TEDDY European Paediatric Medicines Database [346]. For ATMPs compared with conventional drugs, clinical studies on early interventions are at an additional disadvantage, because such studies frequently include long in-patient treatment in an unfamiliar environment and because of the difficulty of achieving truly informed consent for what are often highly sophisticated, hard-to-explain studies with many inherent uncertainties, both known and unknown [347]. Ethical issues generally prevail in pediatric drug studies, but do so even more in ATMP, and specifically GTMP studies. In contrast to other ATMPs, such as CAR-T cells developed to treat aggressive and otherwise lethal cancer types, non-toxic stem cells such as MSCs and autologous, genetically modified HSPCs are often envisioned to treat a variety of debilitating but manageable (if conventional supportive therapy is offered) genetic disorders, which substantially lowers the acceptable risk for corresponding drugs and greatly delays their development process, as well as their evaluation in children [348]. Specifically for HSPC-based GTMPs, concerns about drug-mediated insertional mutagenesis and off-targeting still remain the major hurdles for the endorsement of new pediatric clinical trials of related drugs, whereas for those drugs that make it through clinical trials, the need for the long-term monitoring of recipients for many years after drug administration sets back their final pediatric application approval [349,350,351]. In addition to these considerations for the general acceptability of ATMPs for early treatments, other concerns are uniquely associated with pediatric and/or in utero applications.

#### 7.2.1. Pediatric

The same considerations that drive the long-established gap between the number of adult- and pediatric-approved conventional treatments, such as for small-molecule drugs, also drive the gap between adult and pediatric ATMP applications. For palliative treatments, this creates a dilemma where, in extreme cases, the treating physician may face the choice of unauthorized off-label use of adult medicines to pediatric patients [98], or of leaving the pediatric patient without treatment altogether. For often curative ATMPs, this dilemma is exacerbated, where pediatric patients, as far as has been analyzed, exhibit higher stem cell yields and fewer irreversible disease-related morbidities [194], and would be spared years of palliative treatment and reduced quality of life by early application. On the other side of the argument, safety considerations surrounding experimental treatments for underage patients create a strong counterincentive to trial the participation or approval of pediatric studies. As for adult studies, inclusion criteria for pediatric patients therefore invariably stipulate that trial participants do not respond to standard treatments, although what constitutes a satisfactory response and acceptable quality of life under standard treatments is often open to interpretation. Particularly for slowly progressing diseases, an additional consideration is the ongoing development and prospect of novel and potentially curative treatments, from which younger patients might still benefit later.

#### 7.2.2. In Utero

At present, there is no legal framework for routine in utero ATMP application, chiefly due to the costly and technically demanding nature of the correspondingly limited body of preclinical work, combined with concerns about potential germline transmission and safety to mother and child [352]. Moreover, frequent uncertainty over genotype–phenotype correlation and the actual severity of the disease in postnatal life (e.g., due to genetic modifiers) may not allow clear-cut decisions based on medical necessity, and the relative certainty of a severe in utero or postnatal phenotype is a key criterion for in utero gene therapy according to the consensus statement by the International Fetal Transplantation and Immunology Society [352]. Correspondingly, better-established postnatal treatments are always preferred where disease onset and severity allow, in particular where it is feared that limited studies in large animal models may not have revealed all risks associated with prenatal treatment.

These points may weigh on the mind of bioethics review boards or of the treating physician, but they will also determine attitudes of the affected couples, because for in utero treatment, the notion of parental protection and responsibility is even more acute than for pediatric application. A combined feeling of responsibility, uncertainty over phenotype predictions and over the effectiveness and safety of treatment, and the option of postnatal treatment will combine to create a reluctance by parents to take up in utero therapy, if there are alternatives. After all, the condition might be manageable, or a catastrophic outcome of in utero treatment may come to burden them with the responsibility of having taken a wrong decision. Therefore, the trailblazers for in utero treatments are, and will be, the severest and earliest forms of genetic disease, where the risk–benefit ratio will more readily justify experimental treatments. Here, additional studies in large animal models will be needed to standardize in utero ATMP technologies and to shore up data in support of in utero treatment, as bases for the approval and development of corresponding clinical in utero ATMP applications.

## 8. Perspectives and Conclusions

Recent diagnostic and prognostic advances allow the ever-earlier informed application of ATMP products. More efficient, more affordable therapy is possible by in utero or pediatric applications, with vastly reduced cell and vector requirements for selected ATMP applications. In addition to improving affordability, efficiency and use of GMP resources, early application is fundamental to the treatment of many as-yet untreatable diseases with pre- or perinatal onset. There is, thus, every incentive for ATMPs to narrow the gap for pediatric vs. adult medication, aided by ongoing developments. Be it a growing number of in utero and pediatric studies, prolific research into improved cell isolation and expansion sources and technology, or continuing development of delivery technologies and versatile nanoparticles as vectors, conditions are shifting in favor of ATMPs and for their early application in particular. Critical work remains to address ethical and safety concerns for young or unborn patients, especially where data from adult studies are absent. However, as successful studies accumulate and establishment of underlying technologies and their ethical, regulatory and marketing framework conditions allows further development to gain momentum, prenatal and pediatric application of ATMPs promises safe, efficient and competitive treatments for a growing number of diseases and patients.

Resource, cost, efficiency and suitability advantages, helped by existing regulatory incentives for pediatric and orphan drug development and by a change in attitudes towards advanced therapies, disproportionately favor pediatric and in utero development for ATMPs, which holds the promise of an increase in the proportion of pediatric and early interventions in particular, and correspondingly earlier and better treatments in general.

## Figures and Tables

**Figure 1 pharmaceutics-14-00793-f001:**
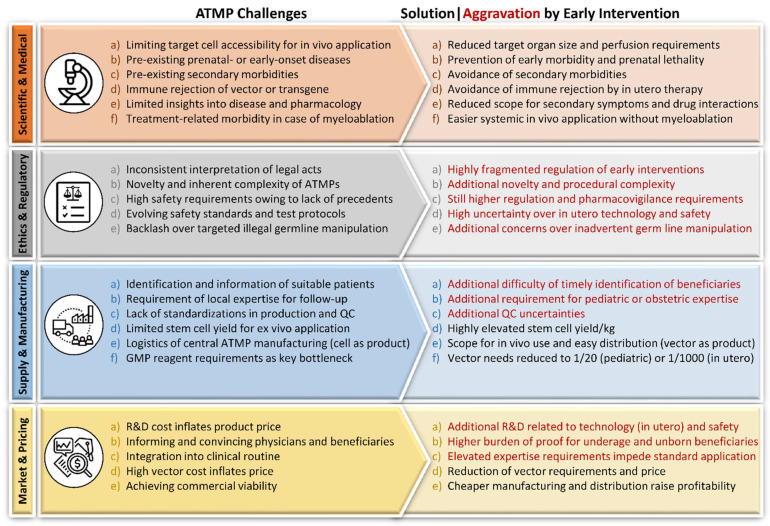
Summary of ATMP challenges and potential aggravation or resolution by early intervention. On the right, aggravating influence by early interventions is shown in red; ameliorating influences are in black.

**Figure 2 pharmaceutics-14-00793-f002:**
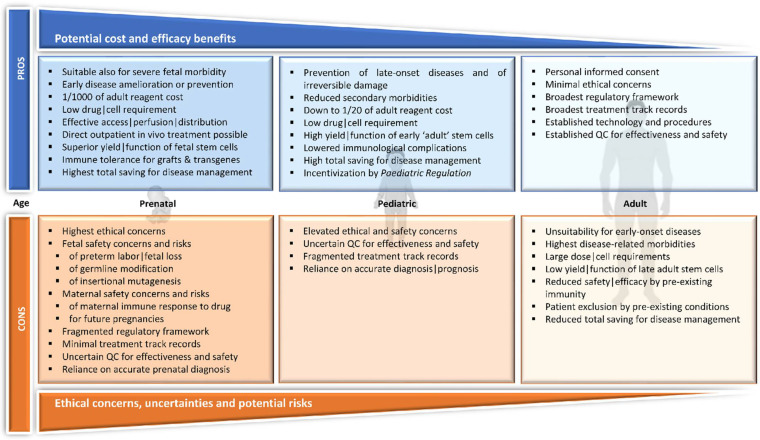
Timeline (prenatal to adult) of ATMP interventions with pros and cons.

**Figure 3 pharmaceutics-14-00793-f003:**
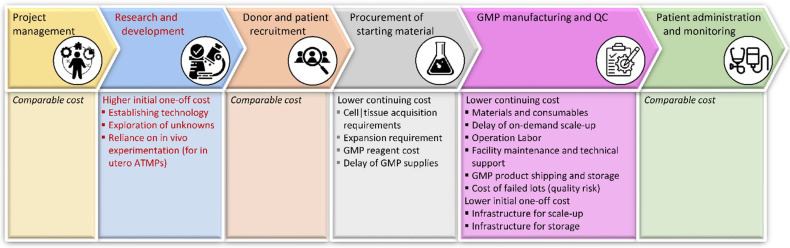
Factors differentially affecting ATMP cost for early vs. adult interventions. Normal black font indicates reduced cost, normal red font indicates increased cost for early interventions.

## Data Availability

Not Applicable.

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
