# Peer review of "Catching Them Early: Framework Parameters and Progress for Prenatal and Childhood Application of Advanced Therapies"

_pharmaceutics, 2022, doi:10.3390/pharmaceutics14040793_

Round 1
Reviewer 1 Report
In this manuscript, Marina and co-workers reviewed the recent progress of advanced therapy medicinal products (ATMPs) for prenatal, perinatal, and pediatric use. This review carries a good summary of existing literature. However, the current MS is hard to follow due to the missing of proper connecting sentences/paragraphs. Please improve the readability of the MS in the resubmission. I recommend a moderate revision for now. The comments are attached below.
- A general introduction is needed before section 1. The author shall explain the history/background using lay language, followed by a quick overview of the paper structure.
- The discussion regarding some figures is insufficient. Please explain and quote the concept presented in the figures more carefully.
- In general, the figures are too wordy. Perhaps the authors could cut the description short and make them more attractive visually. This will allow for better readability.
- For the clinical trials mentioned, please list their NCT number in the paragraph if possible. For example, Line 577 – 581.
- Please remove the unnecessary hyperlinks to some papers (e.g line 386, line 690). Standard citation is fine. It is not usual to mention the full name of a paper in the main manuscript. If you have a specific reason, please explain.
- Outlook/perspective section is missing.
Author Response
In this manuscript, Marina and co-workers reviewed the recent progress of advanced therapy medicinal products (ATMPs) for prenatal, perinatal, and pediatric use. This review carries a good summary of existing literature. However, the current MS is hard to follow due to the missing of proper connecting sentences/paragraphs. Please improve the readability of the MS in the resubmission. I recommend a moderate revision for now. The comments are attached below.
[Response] – We are grateful for the overall favourable assessment. To address the point of readability and beyond the specific changes listed below, sections of this revised version have been re-structured and new sections or short statements introduced to make transitions more intuitive, to explain the choice of content and to provide additional context. We hope that the manuscript is now more accessible to the reader.
1. A general introduction is needed before section 1. The author shall explain the history/background using lay language, followed by a quick overview of the paper structure.
[Response] – Done. We have added a brief “General introduction” that avoids technical terms and gives a summary of content and structure of the manuscript while avoiding substantial overlap with other introductory content. We will be happy to also include a formal table of contents, if that were in the reviewer’s and the reader’s interest.
2. The discussion regarding some figures is insufficient. Please explain and quote the concept presented in the figures more carefully.
[Response] – Done. There is now full correspondence between the chart structure in Figure 1 and sections in the corresponding main text, which has been modified to better reflect the figure content. Figure 3 is now preceded by a corresponding main text section with details that reflect and give background to the chart content. Alone for Figure 2, no more detailed text section is provided, as the figure serves to summarise content of the article across all sections.
3. In general, the figures are too wordy. Perhaps the authors could cut the description short and make them more attractive visually. This will allow for better readability.
[Response] – Done. We have streamlined phrasing and figure design to reduce wordiness and make the figures visually more appealing. Admittedly, the figures are still anything but intuitive diagrams and still have a lot of text content, but any further reduction would have removed from the points made for what is a complex topic. On the plus side, even without reference to the text, the figures can be used in presentations to give a good summary, respectively, of ATMP challenges affected by early intervention, comparisons between fetal, pediatric and adult interventions, and cost factors impacted by early intervention. We hope that the revised figure versions are acceptable.
4. For the clinical trials mentioned, please list their NCT number in the paragraph if possible. For example, Line 577 – 581.
[Response] – Done. NCT numbers were added for the fast majority of clinical trials referred to in the main text, excepting clinical trials with multiple identifiers (e.g. for follow-up studies) and excepting summary statements for a large number of trials.
5. Please remove the unnecessary hyperlinks to some papers (e.g line 386, line 690). Standard citation is fine. It is not usual to mention the full name of a paper in the main manuscript. If you have a specific reason, please explain.
[Response] – Done. Three hyperlinks for references to papers in the same special issue have been removed. The hyperlinks were meant to facilitate recognition of reference to same-issue publications, in case that was of help for editorial writing or that the journal wanted to highlight same-issue papers.
6. Outlook/perspective section is missing.
[Response] – With apologies for deviating from this implicit suggestion of an additional section, to our minds adding an eponymous outlook and perspectives section to an already voluminous review, with much of now sections 6 and 7 already covering outlook and perspectives for early interventions, might not altogether work in the article’s favour. In order to address this point, we have instead renamed the existing “Conclusions” section “Perspectives and conclusions” and in line with that title have added slightly to its already forward-looking content to summarise additional points made in previous sections. We hope that this compromise is acceptable.
Reviewer 2 Report
It was a great pleasure to read this comprehensive Review paper on advanced therapy medicinal products. The paper is nicely organized and well written. The Authors covered almost all important subtopics, from experimental design, and ethical issues to representative clinical trials. Provided illustrations are suitable and helpful. The paper is well/supported by references.
In my opinion, this paper can be accepted in this form.
Author Response
It was a great pleasure to read this comprehensive Review paper on advanced therapy medicinal products. The paper is nicely organized and well written. The Authors covered almost all important subtopics, from experimental design, and ethical issues to representative clinical trials. Provided illustrations are suitable and helpful. The paper is well/supported by references.
In my opinion, this paper can be accepted in this form.
[Response] – We are grateful to the reviewer for this positive assessment and for the time spent going through the manuscript.
Reviewer 3 Report
The review "Catching them early: framework parameters and progress for prenatal and childhood application of advanced therapies” by Carsten W. Lederer et al. is focused on advanced therapy medicinal products (ATMPs) developments for early intervention in prenatal and pediatric stages.
Minor points:
- The authors extensively reviewed the medicinal products of gene (GTMPs) and somatic cell (CTMP) therapy without any reference to tissue engineering products (TEPs); please include some short examples on TEPs.
- Authors should review the Table 1 and its legend. In particular, in the legend they should specify the acronyms effectively used (e.g. A, FA, LAD-1, SCD, MPS-IH, and others) while the superfluous ones (e.g. ATM; NA, etc.) must be eliminated. The reference of the NCT04404595 trial and the type of patients are missing. Furthermore, authors could add in the Table 1 the number of patients including control/reference-treated patients as reported in the legend. Finally, some spaces (e.g. among the CAR-T cells; between ref. 248 and 249-250; between 219 and 254) could be inserted in the Table 1 for better reading.
- Some sentences are not clear (e.g. line 179-182; 883-886); some acronyms (e.g. GT, line 676) could be specified; check line 834 after the reference 301.
Author Response
The review "Catching them early: framework parameters and progress for prenatal and childhood application of advanced therapies” by Carsten W. Lederer et al. is focused on advanced therapy medicinal products (ATMPs) developments for early intervention in prenatal and pediatric stages.
Minor points:
1. The authors extensively reviewed the medicinal products of gene (GTMPs) and somatic cell (CTMP) therapy without any reference to tissue engineering products (TEPs); please include some short examples on TEPs.
[Response] – Done. We have now included key examples for TEP application with corresponding pediatric studies and thank the reviewer for making the article more inclusive of different ATMPs in consequence.
2. Authors should review the Table 1 and its legend. In particular, in the legend they should specify the acronyms effectively used (e.g. A, FA, LAD-1, SCD, MPS-IH, and others) while the superfluous ones (e.g. ATM; NA, etc.) must be eliminated. The reference of the NCT04404595 trial and the type of patients are missing. Furthermore, authors could add in the Table 1 the number of patients including control/reference-treated patients as reported in the legend. Finally, some spaces (e.g. among the CAR-T cells; between ref. 248 and 249-250; between 219 and 254) could be inserted in the Table 1 for better reading.
[Response] – Done. Details on patient numbers were included, the adult-only NCT04404595 trial was removed, the legend was expanded to explain acronyms in the table, surplus acronyms were removed, and borders and spacing were applied to aid readability of the table. If accepted, some of the table formatting will be changed on the journal side towards the proofing stage, but we will try retain as much as possible of the current alignments and formatting.
3. Some sentences are not clear (e.g. line 179-182; 883-886); some acronyms (e.g. GT, line 676) could be specified; check line 834 after the reference 301.
[Response] – Done. The sentences in question and many more throughout the manuscript have been rephrased or simplified to aid understanding. The single appearance of “GT” has been replaced by “gene therapy” and the sentence fragment in original line 834 has been removed. Search for spurious acronyms has further led to the definition of iRGD (which despite single occurrence was retained as a potential search string in its own right).
We thank the Reviewer for the time spent on this manuscript, the suggested improvements and the overall very positive assessment.